# Elevated Mortality Risk in the First Year Post-Diagnosis of Sarcoidosis: A Comprehensive Population-Based Cohort Study

**DOI:** 10.3390/medicina60111787

**Published:** 2024-11-01

**Authors:** Yonatan Shneor Patt, Kassem Sharif, Paula David, Or Hen, Omer Gendelman, Yoav Elizur, Basel Ahmaro, Orly Weinstein, Abdulla Watad, Howard Amital, Niv Ben-Shabat

**Affiliations:** 1Department of Internal Medicine B, Sheba Medical Center, Tel-Hashomer, Ramat Gan 52621, Israel; patt.yonatanshneor@sheba.health.gov.il (Y.S.P.); kassemsharif@gmail.com (K.S.); paulardavid@gmail.com (P.D.); omer.gendelman@sheba.health.gov.il (O.G.); yoav.elizur@mail.huji.ac.il (Y.E.); basel.ahmaro97@hotmail.com (B.A.); watad.abdulla@gmail.com (A.W.); howard.amital@sheba.health.gov.il (H.A.); 2Faculty of Medicine, Tel-Aviv University, Tel-Aviv 69978, Israel; orhen92@gmail.com; 3Department of Gastroenterology, Sheba Medical Center, Tel-Hashomer, Ramat Gan 52621, Israel; 4Department of Internal Medicine C, Sheba Medical Center, Tel-Hashomer, Ramat Gan 52621, Israel; 5Hospital Division, Clalit Health Services, Tel Aviv 67754, Israel; orlyw2@clalit.org.il; 6Department of Health Systems Management, Ben-Gurion University of the Negev, Beer Sheva 84105, Israel; 7NIHR Leeds Musculoskeletal Biomedical Research Unit, Chapel Allerton, Leeds Teaching Hospital Trust, Leeds LS1 3EX, UK

**Keywords:** sarcoidosis, mortality, age, risk factors, epidemiology

## Abstract

*Background and Objectives:* Sarcoidosis, marked by chronic inflammation and granuloma formation, shows a variable clinical course. While many patients have benign outcomes, others face chronic, life-threatening complications. Mortality studies in sarcoidosis show mixed results due to age, ethnicity, sex, and geography, highlighting the need for a comprehensive mortality risk analysis. This study compares mortality risks between sarcoidosis patients and controls, considering demographic and clinical factors, and performs subgroup analyses across different age groups and post-diagnosis periods. *Materials and Methods:* This is a retrospective cohort study that used Clalit Health Services’ electronic database, including patients first diagnosed with sarcoidosis from 2000 to 2016 and age- and sex-matched controls at a 1:5 ratio. Hazard ratios (HR) for all-cause mortality were obtained using the Cox proportional hazard model, adjusted for sociodemographic and clinical variables. *Results:* Sarcoidosis patients showed higher mortality rates (17.7%) than controls (10.6%), with an adjusted HR of 1.79 (95% CI: 1.64–1.96, *p* < 0.001). Subgroup analysis revealed the HR for mortality decreased with age: HR for patients under 50 was 3.04 (95% CI: 2.20–4.21), and for those over 70, it was 1.8 (95% CI: 1.69–2.11). The HR was highest in the first year post-diagnosis. Key mortality predictors included age at diagnosis, male gender, and higher Charlson comorbidity index score. *Conclusions:* Sarcoidosis patients, particularly younger ones and those with higher comorbidity burdens, have elevated mortality risks compared to controls, with the highest HR in the first year post-diagnosis. These findings highlight the most vulnerable period of the disease.

## 1. Introduction

Sarcoidosis, a multisystem granulomatous disease, presents a spectrum of clinical manifestations and outcomes, challenging clinicians and researchers alike. Its etiology remains elusive, contributing to variability in disease progression and prognosis [1]. It is thought to result from an immune response triggered by unidentified antigens in genetically predisposed individuals. The disease involves both the innate and adaptive immune systems, which play a critical role in the formation and maintenance of the granulomas [2]. The clinical trajectory of sarcoidosis ranges from spontaneous resolution, often within two years of diagnosis, to chronic, debilitating, and sometimes fatal courses [3]. Notably, about half of the patients experience self-resolution of the disease within this timeframe, while a significant minority endure persistent or progressive symptoms, leading to substantial morbidity and, in some cases, mortality [4].

Investigations addressing the issue of mortality in sarcoidosis patients are marked by complexities and divergent findings, with some studies suggesting no excess mortality in sarcoidosis patients compared to the general population [5], while others report significantly increased mortality rates [6,7]. Previous research has underscored the impact of demographic factors such as age, ethnicity, and gender on disease outcomes, with higher mortality rates noted in specific subgroups, including African American women aged over 55 [8,9]. Moreover, distinguishable geographical variations are evident in the limited large-scale epidemiological studies on mortality conducted in different regions [10]. 

Specific predictors of mortality in sarcoidosis have been identified in several studies. Comorbidity burden significantly impacts survival [11]. Additionally, sarcoidosis mortality is associated with factors such as age, the extent of lung fibrosis, advanced radiographic stages of the disease, and the presence of pulmonary hypertension [12].

Despite the acknowledged variability in findings, there is a notable lack of comprehensive data, especially regarding the timeline variations of mortality and specific predictors of mortality within the sarcoidosis patient cohort. While previous studies have confirmed an increased risk in sarcoidosis patients for conditions such as malignancy and coronary heart disease, they have not specifically addressed mortality [13,14]. This deficiency highlights the necessity for rigorous, population-based studies. Understanding the factors that influence mortality and overall prognosis in sarcoidosis is crucial for developing effective strategies to prevent complications, optimize treatment approaches, and improve patient outcomes.

In Israel, sarcoidosis presents with unique clinical characteristics, likely shaped by the population’s diverse genetic makeup [15]. However, no epidemiological studies investigating mortality in sarcoidosis have been conducted in this region. By leveraging a large-scale cohort from Israel’s largest Health Maintenance Organization (HMO), our study aims to compare the mortality rates in Israeli sarcoidosis patients and identify the demographic and clinical factors influencing patient outcomes, in relation to the existing literature.

## 2. Materials and Methods

### 2.1. Data Source

The dataset for this investigation was procured from the Clalit Healthcare Services (CHS) comprehensive electronic health records. As Israel’s largest HMO, operating in a payer-provider model, the CHS covers approximately 4.5 million insured members representing diverse ethnic backgrounds. The CHS database continuously draws information from pharmaceutical, medical, and administrative systems covering all insured patients. Employing advanced data-mining techniques, a plethora of patient data is automatically sifted from the database, facilitating meticulous, large-scale epidemiological inquiries on real-time populations. Prior studies have extensively utilized the CHS database [16,17].

### 2.2. Population and Study Design

This retrospective cohort study included all incident cases of sarcoidosis patients initially diagnosed between 1 January 2000 and 31 December 2015. These patients were then matched randomly with controls without a diagnosis of sarcoidosis, in a ratio of 5:1 based on age, sex, and place of residency within a district range. The diagnoses of sarcoidosis were made using ICD codes (ICD-9 code 135 and all affiliated diagnoses) based on hospital discharge letters or physician documentation in primary-care or outpatient visit, with inclusion criteria requiring that the diagnosis code be entered at least twice by two different physicians to strengthen accuracy and reduce the risk of misclassification. The date of the first record marked the diagnosis date. Follow-up for sarcoidosis patients commenced from the date of initial diagnosis (index date), and for controls, from the index date of their matched sarcoidosis patient. Follow-up continued until the earliest of death or the study’s end on 1 November 2016. The mortality data were validated using official death certificates, which are a robust and reliable source of information, ensuring the accuracy and consistency of the data used in our analysis. Since all participants in the cohort were required to maintain their insurance and remain within the HMO until the study’s completion, there was no additional missing data or censoring beyond the end of the follow-up period.

### 2.3. Study Variables

Information available from the CHS database encompassed variables such as age, sex, socioeconomic status (SES), chronic diseases, laboratory results, and date of death for each sarcoidosis patient. SES was delineated based on the poverty index of the member’s residential area, as outlined in the 2008 National Census. Specifically, the poverty index was computed using criteria including household income, educational attainment, marital status, and car ownership, among others. The study population was stratified into three categories according to quartiles (lower 25th percentile, middle 25th to 75th percentile, upper 75th to 100th percentile). BMI was derived from height and weight measurements recorded from the enrollment year until the study period (if available). Diagnoses of comorbidities were extracted from the CHS electronic database and were considered if initially diagnosed before the enrollment date. Comorbidity burden was assessed using the Charlson Comorbidity Index (CCI), a validated tool for predicting mortality in longitudinal studies.

### 2.4. Statistical Analysis

Baseline characteristic differences among various independent variable groups were assessed using *t*-tests or Mann–Whitney U tests for continuous variables, and Pearson’s χ2 test for categorical variables. Survival analysis was conducted employing multivariate Cox proportional hazards modeling. The outcome of interest was all-cause mortality, with the independent variable being sarcoidosis diagnosis. Three models with distinct sets of adjustments were employed: the first univariate, the second adjusting for age, sex, and the CCI, and the third adjusting for age, sex, CCI score, SES, obesity, and smoking. The application of these models was repeated separately in different subgroups based on age, gender, time period from diagnosis, and number of comorbidities. Survival curves were generated utilizing the Kaplan–Meier method, followed by post hoc log-rank comparisons. Initially, we examined the overall study population, followed by a sub-analysis based on gender and age, distinguishing between individuals younger and older than 60 years. Mortality was assessed across three time periods: within one year of sarcoidosis diagnosis, within 1 to 5 years, and over 5 years post-diagnosis. We hypothesized that sarcoidosis may behave more aggressively during the first year, a pattern seen in studies on malignancy and coronary heart disease, justifying these distinct time frames [13,18,19]. Predictors for mortality among sarcoidosis patients were determined using multivariate Cox proportional hazards modeling. Statistical significance was set at *p* < 0.05 for all analyses. A *p*-value less than 0.05 was considered statistically significant. Statistical analyses were performed using the commercial software Statistical Package for the Social Sciences (SPSS for Windows, V.26.0, IBM SPSS Statistics, Armonk, NY, USA).

### 2.5. Ethics

This study received approval from the CHS Ethics Committee in Tel Aviv, Israel, under approval number 0212-17-COM (approval date 1/2018 with an extension to 4/2025). Given the utilization of existing databases, informed consent was not deemed necessary.

## 3. Results

### 3.1. Study Population

The study cohort comprised 3993 patients with sarcoidosis and 19,856 age- and gender-matched controls (Table 1). Both groups shared a median age of 57.1 years (IQR 46–67) and comprised approximately 63% females. Patients with sarcoidosis were more likely to be in a lower SES (41.9% vs. 37.7%; *p* < 0.001) and to be obese (22.5% vs. 16.5%; *p* < 0.001). There were no significant differences in smoking habits between the two groups.

Patients with sarcoidosis exhibited a significantly higher prevalence of chronic comorbidities compared to controls. These included diabetes (23.9% vs. 15.3%; *p* < 0.001), ischemic heart disease (13.3% vs. 9.9%; *p* < 0.001), congestive heart failure (4.2% vs. 1.9%; *p* < 0.001), peripheral vascular disease (2.9% vs. 2.1%; *p* = 0.002), and stroke (4.6% vs. 3.7%; *p* = 0.007). Furthermore, chronic pulmonary diseases were notably more prevalent in the sarcoidosis group (10.8% vs. 4.0%; *p* < 0.001), as were connective tissue diseases (3.2% vs. 1.1%; *p* < 0.001) and chronic kidney disease (4.7% vs. 2.4%; *p* < 0.001). Interestingly, the incidence of malignancy was lower among sarcoidosis patients compared to controls (9.9% vs. 16.3%; *p* < 0.001).

The CCI scores further underscored the greater burden of comorbidities among patients with sarcoidosis. A significantly smaller proportion of sarcoidosis patients had a score of 0 compared to controls (51.8% vs. 66.3%; *p* < 0.001). Higher comorbidity scores across the categories of 1–2, 3–4, 5–6, and >6 were all significantly more prevalent among sarcoidosis patients (*p* < 0.001 for all comparisons).

### 3.2. Overall Mortality and Incidence Rates

Mortality was significantly higher among sarcoidosis patients (17.7%) compared to controls (10.6%). The median follow-up time was 6.8 years for the sarcoidosis group and 7.4 years for controls. The mortality incidence rate per 1000 patient-years was notably higher in sarcoidosis patients at 24.8 (95% CI 23.0–26.7) compared to 13.9 (95% CI 13.4–14.6) in controls. The univariate HR for mortality in sarcoidosis patients was 1.86 (95% CI 1.71–2.02; *p* < 0.001). After adjusting for age, sex, and CCI score (Model 1), and further adjustments for SES, obesity, and smoking (Model 2), the multivariate HR remained significantly elevated at 1.79 (95% CI 1.64–1.95 and 1.64–1.96, respectively; *p* < 0.001).

The analysis by gender revealed significant findings as well. Among men, the mortality rate was 17.2% for sarcoidosis patients versus 10.3% for controls. The multivariate HRs, adjusted for the aforementioned covariates, were 1.78 (95% CI 1.54–2.06) and 1.81 (95% CI 1.56–2.09), respectively (*p* < 0.001). Among women, the mortality rate was 18.0% for sarcoidosis patients compared to 10.8% for controls. The adjusted multivariate HRs were equal to 1.80 (95% CI 1.61–2.01 and 1.71–2.02, respectively; *p* < 0.001). For more information, please see Table 2.

Kaplan–Meier survival curves illustrating the comparative mortality between sarcoidosis patients and healthy controls are presented in Figure 1. The survival analysis revealed a significantly lower 15-year survival rate in the sarcoidosis group, at 62%, compared to 72% among controls. This trend of reduced long-term survival persisted across both genders. Specifically, among males, the survival rate was comparable to the overall cohort analysis (as depicted in Figure 2a). Conversely, the disparity was more marked in females; only 62% of female sarcoidosis patients reached the 15-year survival mark, whereas 74% of female controls did so, highlighting a pronounced gender-specific impact (refer to Figure 2b).

### 3.3. Subgroup Analysis of Mortality Risk

Subgroup analyses demonstrated variability in mortality risks among sarcoidosis patients compared to controls, with stratification by age at diagnosis, time period from diagnosis, and CCI scores. In patients diagnosed before the age of 50, the age-and-sex-adjusted HR for mortality was the highest at 3.04 (95% CI 2.20–4.21; *p* < 0.001), with the multivariate-adjusted HR also elevated at 2.16 (95% CI 1.53–3.03; *p* < 0.001). For those aged 50–59 years and 60–69 years, the mortality risks remained high, though less than the youngest cohort, with multivariate HRs of 1.78 (95% CI 1.41–2.24; *p* < 0.001) and 1.82 (95% CI 1.53–2.17; *p* < 0.001) respectively. The lowest risk was observed in patients aged 70 years and older, with a multivariate HR of 1.69 (95% CI 1.50–1.91; *p* < 0.001).

The mortality risk varied significantly with the time since diagnosis. The highest risk was within the first year of diagnosis, with a multivariate HR of 2.99 (95% CI 2.39–3.75; *p* < 0.001). This risk decreased progressively in later years, with HRs of 1.54 (95% CI 1.34–1.78, *p* < 0.001) for 1–5 years and 1.54 (95% CI 1.35–1.77, *p* < 0.001) for periods extending beyond five years.

Regarding the impact of comorbidities as measured by the CCI, patients with no comorbidities (score 0) exhibited the highest risk for mortality with an HR of 1.95 (95% CI 1.65–2.30; *p* < 0.001). As the comorbidity score increased, the relative mortality risk slightly decreased, with the lowest risk observed for a comorbidity score of 5–6, exhibiting an HR of 1.42 (95% CI 1.09–1.86; *p* < 0.05). For more information, please see Table 3.

### 3.4. Predictors of Mortality Within the Sarcoidosis Cohort

An analysis of mortality predictors within the sarcoidosis cohort identified several significant factors influencing survival outcomes. Age at diagnosis emerged as a prominent predictor, with a univariate HR for every one-year increment of 1.09 (95% CI 1.08–1.09; *p* < 0.001), and a multivariate HR of 1.07 (95% CI 1.06–1.08; *p* < 0.001). Gender also significantly impacted mortality; although the univariate analysis showed no significant difference, the multivariate analysis—adjusted for age and other significant variables—indicated that male gender was associated with an increased risk of mortality, with an HR of 1.31 (95% CI 1.12–1.53; *p* < 0.001).

SES and smoking habits were not significantly associated with mortality in this cohort. However, the CCI proved to be a robust predictor of mortality. Patients with no comorbidities served as the reference group. Increasing comorbidity levels corresponded with progressively higher risks of mortality. Specifically, the HRs for comorbidity scores of 1–2, 3–4, 5–6, and >6 were 2.85 (95% CI 2.34–3.46; *p* < 0.001), 5.28 (95% CI 4.26–6.55; *p* < 0.001), 11.05 (95% CI 8.39–14.55; *p* < 0.001), and 23.03 (95% CI 16.54–32.07; *p* < 0.001), respectively. For more information, please see Table 4.

## 4. Discussion

In this comprehensive population-based study, we observed that individuals diagnosed with sarcoidosis exhibit higher mortality rates compared to the general population. These findings persisted after adjusting for a range of potential confounders and were consistent across various subgroup analyses concerning gender, age, time since diagnosis, and comorbidity profiles. Notably, our study is the first to report that the risk for mortality is most pronounced within the first year following diagnosis. Furthermore, we found that younger patients face the highest risk, which diminishes progressively with increasing age. Upon examining predictors of mortality within the sarcoidosis population, our analysis revealed that older age at diagnosis was associated with increased mortality, as was male gender. 

Understanding mortality in sarcoidosis is crucial for comprehending the disease’s burden, its progression and course, and for informing preventive and therapeutic strategies. While there is a global lack of large-scale epidemiological studies on sarcoidosis mortality, this deficiency is notably pronounced in Israel, where such research is exceptionally sparse. Our study specifically aimed to address this gap by investigating mortality rates within this unique demographic context.

To date, only a few longitudinal studies have investigated and compared mortality in sarcoidosis patients with that of the general population. A 2006 study from the UK and a 2016 study from Sweden both reported excess mortality compared to the general population, with effect sizes comparable to our findings [20,21]. Both of these studies, like ours, utilized a matched cohort and multivariate Cox proportional hazards model, with the Swedish study also adjusting for the CCI. The age and gender composition of the cohorts in these studies were comparable to our cohort. Several additional studies also demonstrated increased mortality in sarcoidosis patients, but they focused on specific patient groups such as black women or patients with a specific phenotype of intrathoracic involvement [6,22]. Additionally, a French study focused on patients with stage IV sarcoidosis, noting a significantly lower ten-year survival rate compared to the general population [7]. Contrary to these findings, a 2015 study from Olmsted County, Minnesota, reported no difference in mortality compared to the general population [23]. Possible explanations for this discrepancy include the small sample size, which may have resulted in insufficient statistical power, the distinct demographic characteristics of the specific population from a single county, and the comparison to standardized population mortality ratios and a comparative cohort.

When comparing the risk in our study with previous research, our HR is higher than the HR of 1.61 found by a Swedish group. They speculated that their relatively lower HR might reflect either improved diagnostics and survival rates in recent decades or particular characteristics of their population, which may possess a better prognosis [21]. Conversely, our findings differ from those of a study by Tukey et al. [6], which reported a much higher HR of 2.4. This substantial variance is likely attributable to their focus on black females, a subgroup known to have a greater susceptibility to sarcoidosis and typically a worse prognosis.

To understand the potential pathophysiological mechanisms related to the observed higher mortality rate in sarcoidosis, it is essential to examine previously reported causes of death. In several studies, the most common cause of death in sarcoidosis patients was the disease itself, rather than its complications. Sarcoidosis was listed as the underlying cause of death in 48% of cases, while neoplasms, cardiovascular diseases, and infections were the leading causes when sarcoidosis was a non-underlying cause of death [24]. These findings indicate that the exaggerated inflammatory response driven by the immune system in sarcoidosis plays a central role in mortality [25].

An interesting finding demonstrated in our study is the pronounced increase in mortality risk within the first year following a sarcoidosis diagnosis. This finding, which for the first time suggests a variation in mortality across the disease timeline, appears to correlate closely with the disease course. Consequently, our results indicate that the initial years post-diagnosis represent the most vulnerable and inflammatory period of the disease. We hypothesize that while most patients experience a self-limited disease course within a few years, those with severe disease presentations suffer from excess comorbidity and mortality immediately following their diagnosis. These patients should be the focus of concentrated medical attention and treatment during the first year after diagnosis. Successfully navigating this period likely leads to a substantially lower mortality risk and an altered disease trajectory. 

Further support for this hypothesis can be seen in a comprehensive study conducted in Sweden investigating the association between various immune-mediated diseases (IMDs), including sarcoidosis, and the risk of subsequent hospitalization for coronary heart disease (CHD) related to coronary atherosclerosis. That study demonstrated an elevated risk of CHD shortly after an IMD diagnosis, with the risk markedly decreasing over time. Specifically, the risk of CHD was highest in the first year post-hospitalization (HR 2.92) and decreased progressively in subsequent years [13,26]. Additionally, several studies have reported a higher risk of malignancies among sarcoidosis patients, particularly in the first years following diagnosis [14,27]. These studies propose that systemic inflammation, particularly pronounced within the first year post-diagnosis, may significantly contribute to the development of atherosclerosis and cancer in sarcoidosis patients. This pattern aligns with our findings, which indicate a peak in mortality risk early in the disease course, thus underscoring a critical and vulnerable period of intense inflammatory activity immediately following the diagnosis.

Although no study has specifically investigated mortality relative to different periods post-sarcoidosis diagnosis, numerous studies have examined mortality through subgroup analyses, focusing on those who received treatment soon after diagnosis. For instance, Rossides et al. [21] examined mortality risks associated with treatment initiation within ±3 months of diagnosis. They found a lower HR (1.13) for patients who did not receive treatment promptly, compared to a significantly higher HR of 2.34 for those who did, suggesting an association between early treatment and more severe disease cases. Similarly, a study from Denmark analyzing 9795 sarcoidosis patients treated with steroids within three years of diagnosis observed a higher mortality rate for treated patients (HR 1.78) compared to those untreated (HR 1.24), further supporting the notion that early treatment may reflect more severe underlying disease [25]. These findings corroborate our conclusions that the initial years post-diagnosis represent a period of increased vulnerability and inflammation, necessitating treatment and correlating with higher mortality rates.

It is important to note that several studies have suggested that sarcoidosis may appear as a paraneoplastic syndrome associated with an underlying malignancy [28,29]. Consequently, the observed increase in mortality within the first year following the diagnosis of sarcoidosis could potentially be linked to an existing malignancy that is diagnosed concurrently with, or leads to the identification of, sarcoidosis.

Given that the vast majority of sarcoidosis cases follow a mild and self-limiting course within the first few years post-diagnosis [30,31], it is critical to determine whether specific subgroups contribute disproportionately to the observed mortality increase. Our study, along with previous research [32], performed subgroup analyses that consistently demonstrated significant mortality risks across various demographics—regardless of gender, age, time since diagnosis, or comorbidity levels. In our study, stratification of sarcoidosis mortality risk according to gender yielded no significant difference between genders. While some previous reports align with our results [21,22], other studies report higher mortality rates among women compared to men [33]. 

An intriguing aspect of our findings is the inverse relationship between the number of comorbidities a patient has, as measured by the CCI, and their HR for mortality. Specifically, patients with a CCI score of 0 exhibited an HR of 1.93, which decreased to 1.39 for those with a score of 5–6. Notably, the absolute number of deaths was lower in patients with fewer comorbidities (6–27%) compared to those with more extensive comorbid conditions (42–44%). Similarly, age-related differences in mortality were pronounced. Older patients exhibited a higher absolute number of deaths (35%) compared to younger patients (2–6%). However, younger patients demonstrated a significantly higher mortality risk compared to matched controls, which gradually decreased with advancing age. This finding mostly signifies that the relative burden of sarcoidosis attributable mortality is lower in older age, with more comorbidities compared to other traditional risk factors, and relative to the overall higher mortality rates. However, another possible contributing factor is the fact that younger patients are often diagnosed with more severe manifestations of sarcoidosis, such as pulmonary fibrosis and cardiac complications, which may contribute to their elevated mortality risk [33,34]. Nonetheless, it provides an important clinical insight and underscores the need for vigilant monitoring and management, even in patients who are young and otherwise healthy. 

Regarding predictors for mortality, our study was the first to consider the CCI as a factor. The CCI is a well-established and widely validated tool across numerous clinical areas [35,36,37]. The results from our study have shown, as expected, higher mortality rates among sarcoidosis patients with increasing age, in males, and with higher comorbidity burden. Though probably not providing any new insights, it reinforces the validity and generalizability of our data. 

While our study boasts several notable strengths, such as its use of a large, extensive, and well-established database, long follow-up period, ethnic diversity, and meticulous methodology, there are significant limitations that must be acknowledged. Firstly, the retrospective nature of our big data study restricted access to detailed clinical information, including specific treatments administered, particularly corticosteroids as well as specific organ involvement. This limitation prevents us from conducting an analysis of mortality risks according to organ involvement, such as cardiac or pulmonary sarcoidosis. Secondly, the diagnosis of sarcoidosis was based solely on coding by specialists without corroborative pathological evidence from biopsies, which could affect diagnostic accuracy. Lastly, in Israel, death certificates do not encompass the cause of death, which limits the understanding of specific mortality factors. However, it is important to note the inherent drawbacks of relying on death certificates that list sarcoidosis as a cause of death. The attribution of death to specific causes in such cases is often problematic, especially for a chronic disease that affects multiple organs. Over time, changes in disease classification and a decrease in the frequency of autopsies have further compromised the reliability of this method. Even when employing death certificates to identify sarcoidosis as a cause of death, the disease was only mentioned in 30% of the cases in the sarcoidosis cohort, underscoring the limitations of this approach [21,38].

## 5. Conclusions

In conclusion, in this large population-based study, we demonstrated higher mortality rates in sarcoidosis patients compared to the general population. Mortality risk was specifically higher early in the course of disease, and in younger patients with less comorbidities. These findings underscore the importance of maintaining a high level of clinical vigilance in sarcoidosis patients during the first year, as this period is the most vulnerable with the highest inflammatory state. We suggest that patients exhibiting a severe disease course in the first year should be treated aggressively with high doses of corticosteroids and other immunosuppressive therapies to mitigate the risk of complications and mortality.

## Figures and Tables

**Figure 1 medicina-60-01787-f001:**
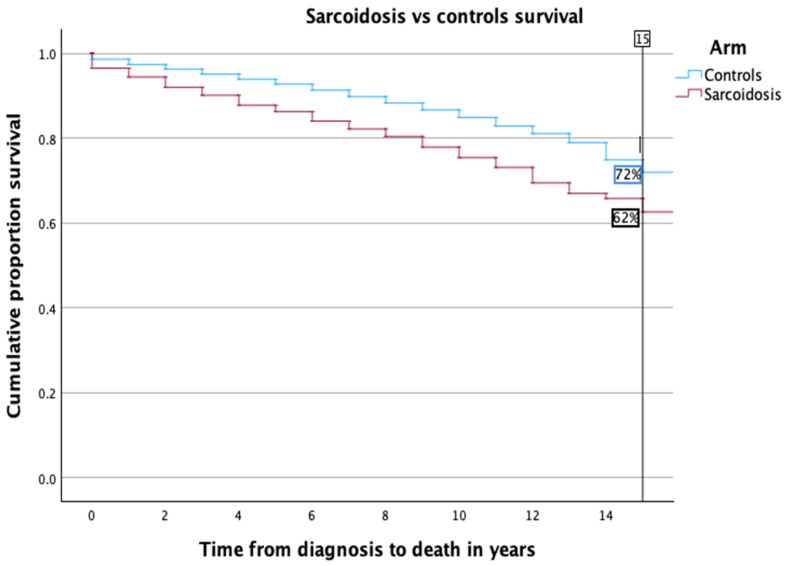
Kaplan–Meier survival curve comparing mortality in sarcoidosis patients vs. healthy controls. This figure illustrates the overall survival rates over time for sarcoidosis patients compared to controls. The graph shows a lower survival rate for sarcoidosis patients, with the difference becoming more pronounced over the 15-year follow-up period. The log-rank test revealed that this difference was statistically significant (*p* < 0.001).

**Figure 2 medicina-60-01787-f002:**
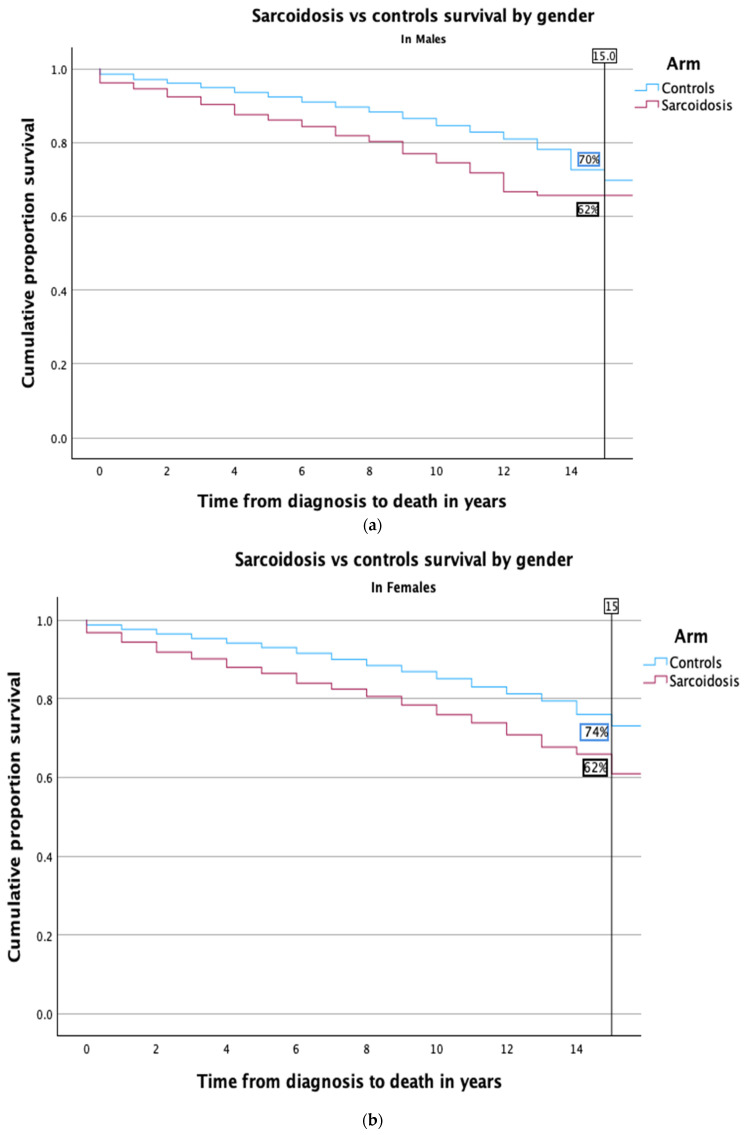
(**a**) Kaplan–Meier survival curve comparing mortality in sarcoidosis patients vs. healthy controls, by male gender. This survival curve specifically compares mortality rates between male sarcoidosis patients and male controls, controlling for gender by stratification. The graph demonstrates a lower survival rate for male sarcoidosis patients compared to their healthy counterparts over the study period. The log-rank test confirmed that this difference was statistically significant (*p* < 0.001). (**b**) Kaplan–Meier survival curve comparing mortality in sarcoidosis patients vs. healthy controls, by female gender. This survival curve compares mortality rates between female sarcoidosis patients and female controls. It demonstrates lower survival rates for female sarcoidosis patients over time. The log-rank test confirmed this difference as statistically significant (*p* < 0.001).

**Table 1 medicina-60-01787-t001:** Baseline characteristics of the study population.

Characteristics	Sarcoidosis (n = 3993)	Controls (n = 19,856)	*p*-Value
Age, median (IQR)	57.1 (46.1–67.1)	57.1 (46.0–67.0)	0.872
Female gender	2522 (63.2%)	12,527 (63.1%)	0.932
Low SES	1654 (41.9%)	7376 (37.7%)	<0.001
Smoking	964 (24.1%)	4985 (25.1%)	0.199
Obesity	897 (22.5%)	3291 (16.5%)	<0.001
Baseline Comorbidities			
Diabetes	953 (23.9%)	3038 (15.3%)	<0.001
Ischemic heart disease	533 (13.3%)	1972 (9.9%)	<0.001
Congestive heart failure	167 (4.2%)	382 (1.9%)	<0.001
Peripheral vascular disease	114 (2.9%)	412 (2.1%)	0.002
Stroke	185 (4.6%)	740 (3.7%)	0.007
Chronic pulmonary disease ^1^	431 (10.8%)	799 (4.0%)	<0.001
Connective tissue disease ^2^	128 (3.2%)	213 (1.1%)	<0.001
Chronic kidney disease	186 (4.7%)	483 (2.4%)	<0.001
Cirrhosis	36 (0.9%)	41 (0.2%)	<0.001
Peptic ulcer disease	287 (7.2%)	1155 (5.8%)	<0.001
Disability	98 (2.5%)	383 (1.9%)	0.031
Dementia	36 (0.9%)	247 (1.2%)	0.068
Malignancy	397 (9.9%)	3596 (16.3%)	<0.001
CCI score			
0	2067 (51.8%)	13,171 (66.3%)	<0.001
1–2	1131 (28.3%)	4398 (22.1%)	<0.001
3–4	530 (13.3%)	1604 (8.1%)	<0.001
5–6	172 (4.3%)	494 (2.5%)	<0.001
>6	93 (2.3%)	189 (1.0%)	<0.001

^1^ including chronic obstructive pulmonary disease, chronic bronchitis, bronchiectasis, interstitial lung disease, and primary pulmonary hypertension. ^2^ including rheumatoid arthritis, systemic lupus erythematous, and systemic sclerosis.

**Table 2 medicina-60-01787-t002:** Mortality rates and risk of sarcoidosis patients and controls.

Group	Variable	Sarcoidosis	Controls
All	Deaths, n/N (%)	710/3993 (17.7)	2121/19,856 (10.6)
	Follow-up time, years, median (IQR)	6.8 (3.6–10.5)	7.4 (4.1–10.9)
	Cumulative patient-years	28,589	151,794
	Mortality incidence rate per 1000 patient’s years (95% confidence Interval)	24.8 (23.0 to 26.7)	13.9 (13.4 to 14.6)
	Univariate HR (95% confidence interval)	1.86 (1.71 to 2.02) *	reference
	Multivariate HR (95% confidence interval), model 1 ^1^	1.79 (1.64 to 1.95) *	reference
	Multivariate HR (95% confidence interval), model 2 ^2^	1.79 (1.64 to 1.96) *	reference
Men	Deaths, n/N (%)	254/1471 (17.2)	761/7329 (10.3)
	Follow-up time, years, median (IQR)	6.4 (3.4–10.0)	6.9 (3.9–10.5)
	Cumulative patient-years	10,074	53,367
	Mortality incidence rate, per 10,000 patient’s years	25.2 (22.2 to 28.5)	14.3 (13.3 to 15.3)
	Univariate HR (95% confidence interval)	1.84 (1.60 to 2.12) *	reference
	Multivariate HR (95% confidence interval), model 1 ^1^	1.78 (1.54 to 2.06) *	reference
	Multivariate HR (95% confidence interval), model 2 ^2^	1.81 (1.56 to 2.09) *	reference
Women	Deaths, n/N (%)	456/2522 (18.0)	1360/12,527 (10.8)
	Follow-up time, years, median (IQR)	7.1 (3.7–10.7)	7.7 (4.3–11.1)
	Cumulative patient-years	18,514	98,426
	Mortality incidence rate, per 10,000 patient-years	24.6 (22.4 to 27.0)	13.8 (13.1 to 14.6)
	Univariate HR (95% confidence interval)	1.87 (1.68 to 2.08) *	reference
	Multivariate HR (95% confidence interval), model 1 ^1^	1.80 (1.61 to 2.01) *	reference
	Multivariate HR (95% confidence interval), model 2 ^2^	1.80 (1.71 to 2.02) *	reference

^1^ Adjusted for age, sex, and Charleson’s comorbidity index score. ^2^ Adjusted for age, sex, CCI score, SES, obesity, and smoking. * *p* < 0.001.

**Table 3 medicina-60-01787-t003:** Mortality risk in sarcoidosis patients compared to controls, a subgroup analysis.

Subgroup	Deaths in Sarcoidosis Patients, n (%)	HR _Age-and-Sex_ (95%CI)	HR _Multivariate_ ^1^ (95%CI)
Age at diagnosis			
<50 years	158 (2.1)	3.04 (2.20 to 4.21) **	2.16 (1.53 to 3.03) **
50–59 years	379 (6.0)	2.13 (1.70 to 2.66) **	1.78 (1.41 to 2.24) **
60–69 years	694 (13.2)	2.22 (1.88 to 2.63) **	1.82 (1.53 to 2.17) **
≥70 years	1600 (34.8)	1.88 (1.69 to 2.11) **	1.69 (1.50 to 1.91) **
Time period from diagnosis of sarcoidosis			
<1 year	138 (3.5)	3.70 (2.98 to 4.61) **	2.99 (2.39 to 3.75) **
1–5 years	289 (7.2)	1.75 (1.53 to 2.00) **	1.54 (1.34 to 1.78) **
≥5 years	283 (7.1)	1.70 (1.49 to 1.94) **	1.54 (1.35 to 1.77) **
CCI score			
0	907 (6.0)	1.93 (1.64 to 2.28) **	1.95 (1.65 to 2.30) **
1–2	909 (6.4)	1.79 (1.54 to 2.09) **	1.79 (1.54 to 2.09) **
3–4	574 (26.9)	1.63 (1.35 to 1.95) **	1.67 (1.38 to 2.01) **
5–6	284 (42.6)	1.39 (1.07 to 1.82) *	1.42 (1.09 to 1.86) *
>6	125 (44.3)	1.57 (1.11 to 2.23) *	1.58 (1.11 to 2.27) **

^1^ Adjusted for age, sex, Charleson’s comorbidity index score, SES, obesity, and smoking. * *p* < 0.05; ** *p* < 0.001.

**Table 4 medicina-60-01787-t004:** Predictors for mortality within the sarcoidosis cohort.

Characteristics	Died During Follow-Up (n = 710)	Alive at Follow-Up (n = 3283)	Univariate HR (95% CI)	*p*	Multivariate ^1^ HR (95% CI)	*p*
Age at diagnosis, median (IQR)	69.9 (61–75)	54.6 (44–63)	1.09 (1.08 to 1.09) ^2^	<0.001	1.07 (1.06 to 1.08) ^2^	<0.001
Male gender	254 (35.8)	1217 (37.1)	1.02 (0.88 to 1.19)	0.768	1.31 (1.12 to 1.53)	<0.0001
Low SES	295 (42.5)	1893 (58.2)	1.09 (0.94 to 1.27)	0.241	-	-
Smoking	136 (19.2)	828 (25.2)	0.96 (0.79 to 1.15)	0.667	-	-
Obesity	153 (21.5)	744 (22.7)	1.33 (1.11 to 1.59)	0.002	0.95 (0.78 to 1.15)	0.593
CCI score					1.26 (1.22 to 1.31)	<0.001
0	177 (24.9)	1890 (57.6)	reference	reference	-	-
1–2	232 (32.7)	899 (27.4)	2.85 (2.34 to 3.46) ^3^	<0.001	-	-
3–4	164 (23.1)	366 (11.1)	5.28 (4.26 to 6.55)	<0.001	-	-
5–6	81 (11.4)	91 (2.8)	11.05 (8.39 to 14.55)	<0.001	-	-
>6	56 (7.9)	37 (1.1)	23.03 (16.54 to 32.07)	<0.001	-	-

^1^ Adjusted for age, gender, and variables that were found statistically significant in the univariate analysis. Enter methos was utilized. Age at diagnosis and CCI score were entered as a continuous variable. ^2^ For every 1-year increment. ^3^ For every 1-index score increment.

## Data Availability

The data used in this study are not available upon request due to the privacy policy of CHS.

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
