# Peer review of "Elevated Mortality Risk in the First Year Post-Diagnosis of Sarcoidosis: A Comprehensive Population-Based Cohort Study"

_medicina, 2024, doi:10.3390/medicina60111787_

Round 1

Reviewer 1 Report

Comments and Suggestions for Authors

Manuscript Title: Elevated Mortality Risk in the First Year Post-Diagnosis of Sarcoidosis: A Comprehensive Population-Based Cohort Study

This is a nice comprehensive cohort study. The manuscript is of interest and is well written.

Overall, it is an interesting article for the readers. All the sections are well outlined and follow an organized format. I have only the following few minor comments for the authors

Comments:

1.     Abbreviations and Acronyms: Ensure that all abbreviations are defined upon initial use and avoid redundant definitions.

2.     In the Introduction Section, specific predictors of mortality within the sarcoidosis should be detailed in a separate para with relevant reference.

3.     Discussion: Section needs to be re-written. It should be crisp, well explanatory, and emphasize the importance of the discussed topic. I suggest a logical flowchart for better understanding of higher mortality rates in sarcoidosis patients in the early course of disease with highest HR in the first year post-diagnosis compared to the younger population with less comorbidities and what factors govern it,

4.     Legend for the figures should be well detailed.

Author Response

Reviewer 1

This is a nice comprehensive cohort study. The manuscript is of interest and is well written.

Overall, it is an interesting article for the readers. All the sections are well outlined and follow an organized format. I have only the following few minor comments for the authors.

Comment 1: Abbreviations and Acronyms: Ensure that all abbreviations are defined upon initial use and avoid redundant definitions.

Response 1: Thank you for your constructive feedback and positive evaluation of the manuscript. The necessary revisions have been made to ensure that all abbreviations are defined upon first use and that redundant definitions are avoided. The following adjustments have been made:

Methods section:

Page 2 - The term "Health Maintenance Organization" was changed to its abbreviation "HMO".

Page 3 - The term “socioeconomic status” was changed to its abbreviation “SES”.

Results section:

Page 4 - The term “socioeconomic status” was changed to its abbreviation “SES”.

Page 4 - The term “hazard ratio” was changed to its abbreviation “HR”.

Page 5 - The term “socioeconomic status” was changed to its abbreviation “SES”.

Discussion:

Page 6 - The term “Charlson Comorbidity Index” was changed to its abbreviation “CCI”.

Table 1-4 (Pages 14-16)

The term “socioeconomic status” was changed to its abbreviation “SES”.

Page 5 - The term “Charlson Comorbidity Index” was changed to its abbreviation “CCI”.

Comment 2: In the Introduction Section, specific predictors of mortality within the sarcoidosis should be detailed in a separate para with relevant reference.

Response 2: Thank you for the suggestion. In accordance with your request, we have added the following paragraph in the introduction with the relevant references:

“Specific predictors of mortality in sarcoidosis have been identified in several studies. Comorbidity burden significantly impacts survival. Additionally, sarcoidosis mortality is associated with factors such as age, the extent of lung fibrosis, advanced radiographic stages of the disease, and the presence of pulmonary hypertension”.

Comment 3: Discussion: Section needs to be re-written. It should be crisp, well explanatory, and emphasize the importance of the discussed topic. I suggest a logical flowchart for better understanding of higher mortality rates in sarcoidosis patients in the early course of disease with highest HR in the first year post-diagnosis compared to the younger population with less comorbidities and what factors govern it.

Response 3: Thank you for the valuable suggestion. In accordance with your request, we have made efforts to revise the discussion section to be more concise and focused, while emphasizing the key findings of our study. The following changes were made:

  • In the second paragraph, beginning with "Understanding mortality in sarcoidosis", we have deleted the sentence "This research gap is particularly relevant given Israel’s ethnically diverse population, which may exhibit distinct clinical manifestations and prognoses of the disease", as this point is already explained in the introduction.
  • In the third paragraph, beginning with "To date, only few longitudinal studies", we shortened the paragraph by reducing the specific details about each study. The revised version is as follows:

"To date, only few longitudinal studies have investigated and compared mortality in sarcoidosis patients with that of the general population. A 2006 study from the UK and a 2016 study from Sweden, both reported excess mortality compared to the general population, with effect sizes comparable to our findings. Both of these studies, like ours, utilized a matched cohort and multivariate Cox proportional hazards model, with the Swedish study also adjusting for the CCI. The age and gender composition of the cohorts in these studies were comparable to our cohort. Several additional studies also demonstrated increased mortality in sarcoidosis patients, but they focused on specific patient groups such as black women or patients with a specific phenotype of intrathoracic involvement. Additionally, a French study focused on patients with stage IV sarcoidosis, noting a significantly lower ten-year survival rate compared to the general population. Contrary to these findings, a 2015 study from Olmsted County, Minnesota, reported no difference in mortality compared to the general population. Possible explanations for this discrepancy include the small sample size, which may have resulted in insufficient statistical power, the distinct demographic characteristics of the specific population from a single county, and the comparison to standardized population mortality ratios and a comparative cohort".

  • In the eighth paragraph, beginning with “Given that the vast majority,” we deleted the following sentences to shorten the discussion and reduce unnecessary detail:

"A closer examination of these divergent findings reveals that studies indicating increased mortality in women often focus on specific demographic groups, particularly the non-Hispanic black population. For instance, research by Mirsaeidi et al. (23) highlighted that sarcoidosis-related mortality rates stratified by sex differed significantly only among African Americans whereas among Caucasians, the difference was relatively minor. As the majority of our cohort are Caucasians it could well explain the difference".

  • To further improve clarity and structure, we have rearranged the discussion to highlight the most important finding—the pronounced increase in mortality risk within the first year following sarcoidosis diagnosis. Thus, we now continue on page 6, after " plays a central role in mortality", with the paragraph beginning, "An interesting finding demonstrated in our study is the pronounced increase in mortality risk within the first year following a sarcoidosis diagnosis".

Comment 4: Legend for the figures should be well detailed.

Response 4: Thank you for your important suggestion. In response, we have added detailed legends for each figure as follows:

Figure 1: This figure illustrates the overall survival rates over time for sarcoidosis patients compared to controls. The graph shows a lower survival rate for sarcoidosis patients, with the difference becoming more pronounced over the 15-year follow-up period. The log-rank test revealed that this difference was statistically significant (p<0.001).

Figure 2a: This survival curve specifically compares mortality rates between male sarcoidosis patients and male controls, controlling for gender by stratification. The graph demonstrates a lower survival rate for male sarcoidosis patients compared to their healthy counterparts over the study period. The log-rank test confirmed that this difference was statistically significant (p<0.001).

Figure 2b: This survival curve compares mortality rates between female sarcoidosis patients and female controls. It demonstrates lower survival rates for female sarcoidosis patients over time. The log-rank test confirmed this difference as statistically significant (p<0.001).

We hope these changes meet your requests and improve the clarity and impact of the discussion.

Reviewer 2 Report

Comments and Suggestions for Authors

The manuscript TitleElevated Mortality Risk in the First Year Post-Diagnosis of Sarcoidosis: A Comprehensive Population-Based Cohort Study” by Yonatan Shneor Patt et al, proposed a large population-based cohort to give a precise examination of mortality risk in patients with sarcoidosis. By addressing the increased mortality risk during the first year following diagnosis, which is very interesting to the reader and researchers in the relevant fields, it significantly advances the subject. Furthermore, the research makes use of a strong database information and suitable statistical methods, such as multivariate Cox proportional hazards models, to ensure that the conclusions are firmly based.

The manuscript is informative, after addressing the few comments given below can fill the relevant gap to enhance the manuscript quality.

Introduction:

1. The introduction does not discuss the significance of the first year following diagnosis, as mentioned in the title, while mentioning overall disease progression and death.

2. The introduction examines generic mortality studies but makes no particular reference to past research on first-year death in sarcoidosis. As it might show the uniqueness of your finding.

Material and methods

3. What is specific criteria to diagnose sarcoidosis to ensure the consistency and reliability of the diagnosis across the cohort.

4. The duration of follow-up is estimated to extend until November 1, 2016, however no information is provided on the median follow-up time or whether any censoring was done for patients who were lost to follow-up.

5. The study does not explain how the time period was classified based on diagnosis or why certain time intervals were chosen for the investigation.

Mortality validation

6. How was the mortality data validated? Was cause of death available and considered in the analysis?

7. Elaborating about cause-specific mortality would be helpful in determining if the higher mortality rate amongst sarcoidosis patients was brought on by the illness itself or any other factors?

 Incorporate a detailed analysis of the mortality risks according to organ involvement to ascertain whether certain organ systems (such as cardiac or pulmonary sarcoidosis) are linked to increased mortality during the initial year following diagnosis or in the long run.

Figure 1.

8. Significance of the Survival Difference in Statistics: Was a log-rank test run to compare the survival distributions of patients with sarcoidosis and controls statistically? For this comparison, what is the p-value?

Discussion:

9. Provide explicit recommendations for clinical management strategies during the first year post-diagnosis.

10. Give particular recommendations on clinical treatment techniques for the first year following diagnosis?

11. Missing Aspect: While comparisons are made with other studies from Sweden, the UK, and the US, there is no discussion of how the death rates in sarcoidosis patients in Israel compare to other areas or nations with distinct healthcare systems and treatment methods.

Author Response

Reviewer 2:

The manuscript Title “Elevated Mortality Risk in the First Year Post-Diagnosis of Sarcoidosis: A Comprehensive Population-Based Cohort Study” by Yonatan Shneor Patt et al, proposed a large population-based cohort to give a precise examination of mortality risk in patients with sarcoidosis. By addressing the increased mortality risk during the first year following diagnosis, which is very interesting to the reader and researchers in the relevant fields, it significantly advances the subject. Furthermore, the research makes use of a strong database information and suitable statistical methods, such as multivariate Cox proportional hazards models, to ensure that the conclusions are firmly based.

The manuscript is informative, after addressing the few comments given below can fill the relevant gap to enhance the manuscript quality.

Introduction:

Comment 1: The introduction does not discuss the significance of the first year following diagnosis, as mentioned in the title, while mentioning overall disease progression and death.

Comment 2: The introduction examines generic mortality studies but makes no particular reference to past research on first-year death in sarcoidosis. As it might show the uniqueness of your finding.

Response 1-2: Thank you for these valuable comments. The reason we did not initially discuss mortality in the first year following diagnosis is that, to our knowledge, no research has specifically analyzed mortality among sarcoidosis patients across several time frames, particularly during the first year post-diagnosis. However, we agree that further discussion regarding the significance of the first year is important. To address this, we have added the following sentence to the Introduction section:

“While previous studies have confirmed an increased risk in sarcoidosis patients for conditions such as malignancy and coronary heart disease, they have not specifically addressed mortality”.

Material and methods

Comment 3: What is specific criteria to diagnose sarcoidosis to ensure the consistency and reliability of the diagnosis across the cohort.

Response 3: Thank you for your comment. Due to the retrospective nature of the study and the large cohort, the diagnoses were made using ICD codes based on hospital discharge letters or physician documentation from primary-care or outpatient visits. To ensure consistency and reliability, the inclusion criteria required that the diagnosis code be entered at least twice by two different physicians to strengthen accuracy and reduce the risk of misclassification.

To clarify this, we have revised the Methods section as follows:

"The diagnoses of sarcoidosis were made using ICD codes based on hospital discharge letters or physician documentation in primary-care or outpatient visits, with inclusion criteria requiring that the diagnosis code be entered at least twice by two different physicians to strengthen accuracy and reduce the risk of misclassification".

Nonetheless, this limitation has been acknowledged in the Discussion section:

“Secondly, the diagnosis of sarcoidosis was based solely on coding by specialists without corroborative pathological evidence from biopsies, which could affect diagnostic accuracy”.

Comment 4: The duration of follow-up is estimated to extend until November 1, 2016, however no information is provided on the median follow-up time or whether any censoring was done for patients who were lost to follow-up.

Response 4: Thank you for your valuable feedback. The median follow-up time is specified in Table 2, with a median duration of 6.8 (3.6-10.5) for sarcoidosis patients and 7.4 (4.1-10.9) for controls. Regarding the issue of censoring, we would like to clarify that this study is a retrospective analysis utilizing electronic health records (EHR) to investigate mortality in sarcoidosis patients. The primary source of censoring, apart from reaching the end of the study follow-up period, would be if a patient left the HMO. However, since all participants in the cohort were required to maintain their insurance and remain within the HMO until the study's completion, there is effectively no additional missing data or censoring beyond the end of the follow-up period. To address your comment and improve clarity, we have added the following segment to the Methods section:

Since all participants in the cohort were required to maintain their insurance and remain within the HMO until the study's completion, there was no additional missing data or censoring beyond the end of the follow-up period”.

Comment 5: The study does not explain how the time period was classified based on diagnosis or why certain time intervals were chosen for the investigation.

Response 5:

Thank you for your insightful comment. We agree that the classification of time periods was not clearly explained in the manuscript. Although the time frames of mortality incidence following sarcoidosis diagnosis have not been extensively explored in the literature, we hypothesized that the disease may behave more aggressively during the first year post-diagnosis. This assumption is supported by several studies investigating other conditions. For example, research on the prevalence of malignancy and coronary heart disease found a significantly higher association during the first year following diagnosis. Based on these findings, we chose to examine mortality in three distinct time frames: the first year post-diagnosis, the medium term (1-5 years), and the long term (>5 years).

To clarify this in the manuscript, we have added the following text to the methods section:

"Mortality was assessed across three time periods: within one year of sarcoidosis diagnosis, 1-5 years, and over 5 years post-diagnosis. We hypothesized that sarcoidosis may behave more aggressively during the first year, a pattern seen in studies on malignancy and coronary heart disease, justifying these distinct time frames".

Mortality validation

Comment 6: How was the mortality data validated? Was cause of death available and considered in the analysis?

Response 6:

Thank you for your important question. The mortality data was validated using death certificates, which are highly reliable and consistent sources of information, thus requiring no further validation. To clarify this point, we have added the following sentence to the Methods section:

"The mortality data was validated using official death certificates, which are a robust and reliable source of information, ensuring the accuracy and consistency of the data used in our analysis".

Regarding the availability and consideration of cause of death, we will address this in more detail in the response to the next comment.

Comment 7: Elaborating about cause-specific mortality would be helpful in determining if the higher mortality rate amongst sarcoidosis patients was brought on by the illness itself or any other factors?

Incorporate a detailed analysis of the mortality risks according to organ involvement to ascertain whether certain organ systems (such as cardiac or pulmonary sarcoidosis) are linked to increased mortality during the initial year following diagnosis or in the long run.

Response 7:

Thank you for your insightful comment. Unfortunately, in Israel, death certificates do not include the specific cause of death, which prevents us from directly investigating the reasons for the higher mortality observed among sarcoidosis patients. However, recognizing the importance of this question, we have referenced other studies in the discussion section that have investigated causes of death in sarcoidosis:

"To understand the potential pathophysiological mechanisms related to the observed higher mortality rate in sarcoidosis, it is essential to examine previously reported causes of death. In several studies, the most common cause of death in sarcoidosis patients was the disease itself, rather than its complications. Sarcoidosis was listed as the underlying cause of death in 48% of cases, while neoplasms, cardiovascular diseases, and infections were the leading causes when sarcoidosis was a non-underlying cause of death. These findings indicate that the exaggerated inflammatory response driven by the immune system in sarcoidosis plays a central role in mortality".

Additionally, we have addressed this limitation in the discussion section:

"Lastly, in Israel, death certificates do not encompass the cause of death, which limits the understanding of specific mortality factors".

Regarding your question about conducting a detailed analysis of mortality risks according to organ involvement - due to the retrospective nature of this study and the large cohort, the diagnoses were made using ICD codes, and unfortunately, we do not have detailed clinical characteristics for each patient, including the severity of sarcoidosis or specific organ involvement. To address this limitation, we have added the following to the limitation section:

“Firstly, the retrospective nature of our big data study restricted access to detailed clinical information, including specific treatments administered, particularly corticosteroids as well as specific organ involvement. This limitation prevents us from conducting an analysis of mortality risks according to organ involvement, such as cardiac or pulmonary sarcoidosis”.

Figure 1.

Comment 8: Significance of the Survival Difference in Statistics: Was a log-rank test run to compare the survival distributions of patients with sarcoidosis and controls statistically? For this comparison, what is the p-value?

Response 8: Thank you for your insightful query. In response, we have included the results of the log-rank test used to compare the survival distributions of sarcoidosis patients and controls. The p-value for this comparison has been provided in the detailed legends for each figure, as follows:

Figure 1: This figure illustrates the overall survival rates over time for sarcoidosis patients compared to controls. The graph shows a lower survival rate for sarcoidosis patients, with the difference becoming more pronounced over the 15-year follow-up period. The log-rank test revealed that this difference was statistically significant (p<0.001).

Figure 2a: This survival curve specifically compares mortality rates between male sarcoidosis patients and male controls, controlling for gender by stratification. The graph demonstrates a lower survival rate for male sarcoidosis patients compared to their healthy counterparts over the study period. The log-rank test confirmed that this difference was statistically significant (p<0.001).

Figure 2b: This survival curve compares mortality rates between female sarcoidosis patients and female controls. It demonstrates lower survival rates for female sarcoidosis patients over time. The log-rank test confirmed this difference as statistically significant (p<0.001).

Discussion:

Comment 9: Provide explicit recommendations for clinical management strategies during the first year post-diagnosis.

Comment 10: Give particular recommendations on clinical treatment techniques for the first year following diagnosis?

Response 9-10:

In accordance with your recommendations, we have expanded the conclusion section as follows:

"These findings underscore the importance of maintaining a high level of clinical vigilance in sarcoidosis patients during the first year, as this period is the most vulnerable with the highest inflammatory state. We suggest that patients exhibiting a severe disease course in the first year should be treated aggressively with high doses of corticosteroids and other immunosuppressive therapies to mitigate the risk of complications and mortality".

Comment 11: Missing Aspect: While comparisons are made with other studies from Sweden, the UK, and the US, there is no discussion of how the death rates in sarcoidosis patients in Israel compare to other areas or nations with distinct healthcare systems and treatment methods.

Response 11: Thank you for your valuable comment. In response, we have added a paragraph comparing the mortality rates in Israel with those reported in other countries:

When comparing the risk in our study with previous research, our HR is higher than the HR of 1.61 found by a Swedish group. They speculated that their relatively lower HR might reflect either improved diagnostics and survival rates in recent decades or particular characteristics of their population, which may possess a better prognosis. Conversely, our findings differ from those of a study by Tukey et al., which reported a much higher HR of 2.4. This substantial variance is likely attributable to their focus on black females, a subgroup known to have a greater susceptibility to sarcoidosis and typically a worse prognosis”.

We believe these revisions address your comments and improve the clarity and impact of the discussion.

Reviewer 3 Report

Comments and Suggestions for Authors

The current study compared mortality risks between sarcoidosis patients and controls, considering demographic and clinical factors, and performed subgroup analyses across different age groups and post-diagnosis periods. The study included a large sample size.

However, there are some minor remarks:

-        The pathophysiological mechanisms of sarcoidosis need to be mentioned in the Introduction section.

-        The Discussion section should be extended with more information explaining the potential pathophysiological mechanisms related the obtained results.

-        A „p“ level of significance needs to be explained in the Statistical analysis subsection.

-        The abbreviations should be uniform, e.g. „P“ or „p“.

-        The abbreviations when first introduced should be used consistently thereafter, e.g. „Charlson Comorbidity Index”.

-      

Comments on the Quality of English Language

-        English proof editing is recommended.

Author Response

Reviewer 3:

The current study compared mortality risks between sarcoidosis patients and controls, considering demographic and clinical factors, and performed subgroup analyses across different age groups and post-diagnosis periods. The study included a large sample size.

However, there are some minor remarks:

Comment 1: The pathophysiological mechanisms of sarcoidosis need to be mentioned in the Introduction section.

Response 1: Thank you for your valuable suggestion. In accordance with your request, we have added the following sentences to the introduction with the relevant reference:

“It is thought to result from an immune response triggered by unidentified antigens in genetically predisposed individuals. The disease involves both the innate and adaptive immune systems, which play a critical role in the formation and maintenance of the granulomas”.

Comment 2: The Discussion section should be extended with more information explaining the potential pathophysiological mechanisms related the obtained results.

Response 2: Thank you for your insightful suggestion. In accordance with your request, we have added the following paragraph to the discussion section, emphasizing the potential pathophysiological mechanisms underlying the higher mortality observed in our study:

"To understand the potential pathophysiological mechanisms related to the observed higher mortality rate in sarcoidosis, it is essential to examine previously reported causes of death. In several studies, the most common cause of death in sarcoidosis patients was the disease itself, rather than its complications. Sarcoidosis was listed as the underlying cause of death in 48% of cases, while neoplasms, cardiovascular diseases, and infections were the leading causes when sarcoidosis was a non-underlying cause of death. These findings indicate that the exaggerated inflammatory response driven by the immune system in sarcoidosis plays a central role in mortality".

Furthermore, to explore the reason behind the higher mortality rate within the first year, we have formulated a hypothesis and supported it with evidence from other studies (p 6-7):

"We hypothesize that while most patients experience a self-limited disease course within a few years, those with severe disease presentations suffer from excess comorbidity and mortality immediately following their diagnosis... This pattern aligns with our findings, which indicate a peak in mortality risk early in the disease course, thus underscoring a critical and vulnerable period of intense inflammatory activity immediately following the diagnosis".

Comment 3: A „p“ level of significance needs to be explained in the Statistical analysis subsection.

Response 3: We agree that the p-level of significance should be clearly defined. We have added the following explanation to the Statistical analysis section:

"Statistical significance was set at p < 0.05 for all analyses. A p-value less than 0.05 was considered statistically significant".

Comment 4: The abbreviations should be uniform, e.g. „P“ or „p“.

Response 4: Thank you for your comment. We have revised the manuscript and changed all instances of "P" to "p" to ensure uniformity throughout the text.

Comment 5: The abbreviations when first introduced should be used consistently thereafter, e.g. „Charlson Comorbidity Index”.

Response 5: The necessary revisions have been made to ensure that all abbreviations are defined upon first use and that redundant definitions are avoided. The following adjustments have been made:

Methods section:

Page 2 - The term "Health Maintenance Organization" was changed to its abbreviation "HMO".

Page 3 - The term “socioeconomic status” was changed to its abbreviation “SES”.

Results section:

Page 4 - The term “socioeconomic status” was changed to its abbreviation “SES”.

Page 4 - The term “hazard ratio” was changed to its abbreviation “HR”.

Page 5 - The term “socioeconomic status” was changed to its abbreviation “SES”.

Discussion:

Page 6 - The term “Charlson Comorbidity Index” was changed to its abbreviation “CCI”.

Table 1-4 (Pages 14-16)

The term “socioeconomic status” was changed to its abbreviation “SES”.

Page 5 - The term “Charlson Comorbidity Index” was changed to its abbreviation “CCI”.

We aim that these revisions address your comments and contribute to the clarity and overall impact of the manuscript.
